# Behavioral Changes, Adaptation, and Supports among Indonesian Female Sex Workers Facing Dual Risk of COVID-19 and HIV in a Pandemic

**DOI:** 10.3390/ijerph19031361

**Published:** 2022-01-26

**Authors:** Gede Benny Setia Wirawan, Brigitta Dhyah K. Wardhani, Putu Erma Pradnyani, Afriana Nurhalina, Nurjannah Sulaiman, Evi Sukmaningrum, Luh Putu Lila Wulandari, Pande Putu Januraga

**Affiliations:** 1Center for Public Health Innovation, Faculty of Medicine, Udayana University, Denpasar 80113, Indonesia; benny.wirawan007@gmail.com (G.B.S.W.); gittadhyah@gmail.com (B.D.K.W.); pradnyanierma@gmail.com (P.E.P.); 2Ministry of Health, Jakarta 12950, Indonesia; afriana4272@gmail.com (A.N.); aneukinong@yahoo.com (N.S.); 3Faculty of Psychology, Atma Jaya Catholic University of Indonesia, Jakarta 12930, Indonesia; evi.sukmaningrum@gmail.com; 4Kirby Institute, University of New South Wales, Sydney 2033, Australia; putuwulandari@gmail.com; 5Department of Public Health and Preventive Medicine, Faculty of Medicine, Udayana University, Denpasar 80113, Indonesia

**Keywords:** female sex workers, COVID-19 pandemic, social support, HIV prevention, condom, online sex work, Indonesia

## Abstract

The objective of this study is to explore the impacts of COVID-19 and changes taking place among the Indonesian female sex worker (FSW) community during the COVID-19 pandemic and the predictors of these changes. We conducted a cross-sectional online survey and selected the participants using a purposive snowball sampling technique. Incentives were provided to participants in the form of a 5 USD e-wallet balance. Variables of interest included adaptation to online sex work, adherence to COVID-19 prevention measures during sex work, number of clients, income reduction, social support, condom access, and condom use frequency. Sociodemographic data and COVID-19 fear index values were also collected. Final analysis included 951 FSWs, of whom 36.4% of had adapted to online sex work and 48.6% had practiced COVID-19 prevention measures. Major reductions in client frequency and income were reported by 67.8% and 71.1% of respondents, respectively. However, only 36.3% of FSWs reported they had ever received any form of social support from any parties, public or private. Meanwhile, 16.7% encountered difficulties in accessing condoms and 12.5% reported less frequent condom use during the pandemic. Easy access to condoms was the main factor influencing the frequency of condom use. As expected, staying in employment protected FSWs from major income loss, while education and younger age predicted adaptive behavioral changes, such as taking up online sex work. The COVID-19 pandemic has disrupted access to socioeconomic support systems and HIV prevention services among FSWs and has further exposed them to the dual jeopardy of HIV and COVID-19 infections.

## 1. Introduction

The impacts of the COVID-19 pandemic have been seen in all layers of society. In September 2021, the caseload was over 224 million, and mortalities were more than 4.6 million globally [1]. Despite the rollout of vaccines in early 2021, a record-high number of new cases was reported in January 2022 due to the outbreak of the new Omicron variant [2]. The constant change in case numbers has led to the sporadic imposition of social restrictions, especially in hard-hit regions, in addition to the vaccine rollout.

Female sex workers (FSWs) are especially vulnerable to the impacts of the COVID-19 pandemic, exacerbating the already high health risks their engagement in sex work activities pose. Their vulnerability stems from various and interconnected economic, sociocultural, and demographic factors. The close physical contact required of FSWs during sex service increases their likelihood of contracting COVID-19 from clients. At the same time, social restrictions may have led to reduced numbers of clients and thus reduced incomes for FSWs. In response to economic insecurity, FSWs may be likely to pursue other means, including engaging in riskier behaviors, related to both COVID-19 and HIV risks, simply to keep themselves in business. For example, in a bid to attract more clients and thus more income, many FSWs have been forgoing the use of condoms [3]. A similar logic could also apply during the COVID-19 pandemic, whereby to keep their head above water, FSWs may forgo COVID-19-preventive behaviors during their transactional sex activities, exposing them to the heightened risk of COVID-19 transmission [4]. 

Meanwhile, sociocultural and demographic factors also intersect to compound the economic and health impacts of the pandemic on this group. As a marginalized community in the overwhelmingly conservative country of Indonesia, FSWs face difficulties in accessing government supports. While known sex workers may also face ostracism from their own community of friends and family [5,6], demographic data also show that many FSWs were internal migrants who often had no social support system in their current place of residence. This would exacerbate the difficulties for FSWs to access support, worsening their economic insecurities and, in turn, prompting risky behaviors [7,8].

HIV risks have been further heightened as the healthcare system continues to be stretched by the demands of the COVID-19 pandemic. As part of the HIV mitigation program, local health authorities have been providing free access to condoms, sterile needles, and antiretroviral drugs, along with conducting mobile HIV tests in FSW-heavy areas. However, the pandemic has drained much of the financial and human resources from these programs, leaving key populations, including FSWs, at increased risk of HIV infection [9,10,11]. 

There have been recommendations on the need to focus on providing services and supports to the affected FSWs as well as guidance to mitigate their COVID-19 and HIV risks during the pandemic. It should be noted that safe engagement in sex work may be achievable through the adoption of two behavioral changes: online sex work and strict adherence to COVID-19 preventive measures during in-person sex work [12,13,14]. Adoption of strict procedures to limit droplet exchange during the sexual encounter with clients can mitigate COVID-19 transmission risk for FSWs engaging in in-person sex work [15,16]. The disbanding of brothels or other forms of localizations during the pandemic is not recommended, in order to allow FSWs to continue to have access to health services, including HIV prevention services [12,17]. While brothels do not themselves provide HIV prevention services, their role as hotspots or venues facilitates access to both publicly and privately run HIV prevention services, such as condom distribution and mobile HIV testing targeting FSWs, and is performed in many settings [7,8].

Online sex work has offered an especially promising way for FSWs to adapt to life during the pandemic. In fact, such a shift had been taking place even before the onset of the pandemic [18,19]. It is easy to envision how the increasing engagement with the Internet during the pandemic would accelerate shift to online transactions. It is a move that may help FSWs adapt and be financially productive during the pandemic while staying safe from COVID-19 [12]. Several studies have noted the association of online sex work with better HIV risk mitigation [20,21]. 

Notwithstanding these recommendations, recent qualitative data showed that the pandemic has affected FSW communities around the world. Even with the adoption of these behavioral changes, i.e., online sex work and strict adherence to COVID-19 preventive measures during in-person sex work, many still reported severe socioeconomic impacts resulting from difficulties in accessing preventive aids and other resources [4,22,23]. FSWs have also faced higher barriers to accessing healthcare [24], including trouble accessing condoms and HIV tests [9].

Despite the previous qualitative reporting of impacts and changes affecting FSWs during the COVID-19 pandemic, very few quantitative data contributing to a description of the overall situation are available. These data are germane to understanding the situation in depth and informing future responses to any potential disruptions to service targeting this community as a result of the current and any future pandemic.

Thus, this study was conducted primarily to identify the behavioral changes, impacts, and social supports operating among the Indonesian FSWs to mitigate COVID-19 and HIV risks during the pandemic. The secondary objective was to elucidate factors affecting socioeconomic impacts and behavioral changes in response to COVID-19 pandemic among the Indonesian FSWs. 

## 2. Materials and Methods

### 2.1. Study Settings

An online survey was run from 30 September to 31 October 2020 among FSWs in four Indonesian metropolitan areas: Greater Jakarta, Bandung, Yogyakarta, and Bali. These cities were selected for their large concentrations of HIV cases and members of the key study population [25]. At the same time, these metropolitan areas had suffered the brunt of COVID-19 cases in the early phases of the pandemic in Indonesia [26]. Specifically, by the end of data collection on 31 October 2020, the provinces where these metropolitan areas are located recorded over 158,000 cumulative COVID-19 cases, nearly 40% of the national figure of 410,000 cases. Consequently, these areas were among the targets of Indonesian social restriction measures aimed to limit COVID-19 transmissions [10,27]. 

The combined effects of the pandemic and related responses had led to significant socioeconomic and health impacts in these areas. Overall, Indonesia had suffered a recession that only subsided in the second quarter of 2021. One of the study areas, Bali, was also one of the most heavily impacted areas economically, given that it is highly dependent on the tourism industry [28]. The socioeconomic impact had also severely affected migrant populations which, in these metropolitan areas, include a significant portion of FSWs, who may lack the social support available to the local native population [9]. Meanwhile, the Indonesian government had adapted HIV prevention services, including adopting alternative counseling procedures and condom distributions. It should be noted, however, that these measures may not be equally accessible to all layers of key population communities [29]. 

### 2.2. Study Design

This study employed a quantitative analytic design to describe socioeconomic and behavioral changes among FSWs related to COVID-19 and HIV risks during the pandemic, as well as the factors influencing these changes. Data were collected using a self-administered online survey developed in the Indonesian language. It consisted of around 200–250 multiple choice items with branching logics based on participants’ answers; completion by FSWs required around 24 to 58 min. 

Target participants were FSWs located in the four metropolitan areas in Indonesia, selected by snowball sampling and network sampling techniques. We collaborated with 15 local community-based organizations to distribute the survey recruitment materials among FSW communities in the target areas, which included the four metropolitan areas of Jakarta, Bandung, Yogyakarta, and Bali. Collaborating organizations utilized their social media and networks to distribute the recruitment materials which, included digital pamphlets and URLs for the self-administered online survey. These organizations also facilitated access to the survey for those potential participants who had difficulties in accessing the survey. Incentives for respondents were provided in the form of e-wallet balance amounting to IDR 75,000 (equal to around 5 USD).

We collected data on the impacts of behavioral changes on client frequency as perceived by respondents. Regarding behavioral changes in response to the pandemic, we collected data on the shift to online sex services and adoption of COVID-19-preventive practices for in-person sex work. We also collected data on changes to HIV-related indicators, including changes to condom access and condom use frequency during the pandemic.

Shift to online sex work during the pandemic was measured in dichotomous categories including services such as chat sex/sexting, call sex, and video call sex with paying clients. It should be noted that the adult content industry is not mainstream in Indonesia, even among FSWs, and accessible only to the most Internet-savvy FSWs. An FSW was considered to engage in online sex work if she indicated she had been providing at least one of these services.

Adoption of COVID-19-prevention measures during sex work was based on an 9-item list of behaviors recommended by local FSWs community organizations. Adherence to each recommended behavior was scored using a Likert scale ranging from “never” (score 0) to “always” (score 3). Factor analysis was conducted to assess the construct validity and to determine the number of factors to retain. The results, available in Appendix A, show two significant subscales: (1) sanitation and personal hygiene, consisting of 6 items, as well as (2) use of personal protective equipment (PPE) during sex work, consisting of 3 items. One item (i.e., mask-wearing behavior) was found to be cross-loading but grouped together with the PPE usage subscale based on the strength of the loading factor. Internal reliability analysis shows that both subscales were internally reliable with a Cronbach *α* value of 0.915 and 0.842. The score was then dichotomized using the median value (72.22% for sanitation and personal hygiene, 33.33% for the PPE use) as the cutoff point into high-adherence (i.e., higher than or equal to the median value) and low-adherence (lower than the median value) groups.

We also collected data on factors that may have affected the adoption of behavioral changes and the level of impacts on client frequency changes. These factors included sociodemographic characteristics, HIV status, and fear of COVID-19. Sociodemographic variables collected included age, education level, employment outside of sex work, marital status, and living situation. Meanwhile, fear of COVID-19 was measured using an adaptation of a previously validated instrument [30]. The score was dichotomized using the median as the cut-off point into high- and low-fear groups.

### 2.3. Statistical Analysis

We conducted univariate and multivariate analyses to understand factors affecting behavioral adoption of online sex work during the pandemic and adherence to COVID-19-preventive behaviors. Univariate and multivariate analysis was conducted to understand how these behavioral changes affected client frequency. All analysis was conducted using binomial logistic regression with effect size presented as crude odds ratio (cOR) for univariate analysis and adjusted odds ratio (aOR) for multivariate analysis results, respectively. Only variables with *p* < 0.25 in the univariate logistic regression were included in the multivariate regression. All analyses were performed on IBM SPSS 23.0 for Microsoft Windows (IBM Corp, Armonk, NY, USA).

### 2.4. Ethical Considerations

All respondents participating in this study received an information package regarding the study and consented to participate. The protocol for this study was reviewed and approved by the Human Research Ethics Committee of the Faculty of Medicine, Udayana University/Sanglah Hospital with the ethical clearance number of 1806/UN14.2.2.VII.14/LT/2020.

## 3. Results

Our survey garnered 1266 responses. After the cleaning process and removal of duplicate entries, missing values, and invalid responses, the final analysis included 951 FSWs, whose details are presented in Table 1. The distribution of this figure among the four metropolitan areas was as follows: 334 (35.1%) from the Greater Jakarta area, 250 (26.3%) from Bandung, 152 (16.0%) from Yogyakarta, and 215 (22.6%) from Bali. Age of respondents was non-normally distributed, with the median age of 26 years and interquartile range between 23 and 31 years. Education level was low—only a little over 50% had completed high school. Most FSWs (63.3%) also reported no other employment outside of sex work. The majority of FSWs were not currently married, with 40.8% being single and 38.5% widowed or divorced. However, 41.6% of FSWs reported currently living with family. As many as 9.1% of FSWs reported having an HIV-positive status. With respect to COVID-19, 48.2% of FSWs indicated they were fearful of it.

During the pandemic, some FSWs reported behavioral changes related to the pandemic situation. A score was calculated for each subscale, presented as percentage points. Descriptive analysis of the scores showed that the mean score of sanitation and personal hygiene was 73.22% ± 25.23%, and the mean score of the PPE use was 41.20% ± 32.51%. The variance value for each subscale was 6.4% and 10.6%, respectively. 

Changes also occur in the form of online sex work. A little over a quarter (26.4%) reported taking up online sex work in various forms, including chat sex, call sex, and video call sex services. Meanwhile, the impact of the pandemic was reflected in reduced client frequency and income as reported by the majority of FSWs. However, only 36.3% of FSWs reported that they had ever received any form of social support from any parties, public or private. An impact was also felt concerning HIV prevention. Increased difficulties in accessing condoms during the pandemic were reported by 16.9% of FSWs, while 12.5% reported reduced condom use altogether. 

Table 2 and Table 3 show that the pandemic-related behavioral changes reported by FSWs were mostly motivated by fear of contracting COVID-19. A high COVID-19 fear index score was associated with taking up online sex services with an aOR of 1.69 (95% CI 11.27–2.25). This was associated with both forms of COVID-19 prevention measures during sex work, with an aOR of 1.44 (95% CI 1.10–1.89) for high adherence to sanitation and personal hygiene and aOR of 1.40 (95% CI 1.07–1.83) for adhering to PPE use during sex work. Taking up online sex work as an adaptive behavior change during the pandemic was determined by various factors, although there were different patterns of determinants for each COVID-19 prevention subscale. Adherence to sanitation and personal hygiene was associated with location and HIV status while the PPE use was associated with education.

Meanwhile, the pandemic’s impact on income (Table 4) also appears to have been partially self-motivated. As expected, a major reduction in client frequency translated into major reduction in income, with an aOR of 1.88 (95% CI 1.36–2.59). Furthermore, income reduction was associated with a higher COVID-19 fear index score with an aOR of 3.32 (95% CI 2.41–4.56), which indicated the involvement of some self-motivation in client reduction resulting in loss of income. The risk of more severe socioeconomic impacts was also associated with demographic and socioeconomic backgrounds. Predictably, employment other than sex work provided some protection against income reduction. Meanwhile, FSWs living in areas with high tourism dependency prior to the pandemic, such as Yogyakarta, were more likely to be subject to the impact of income reduction. Cluster analysis of different types of online sex work as determinants also showed that engagement in video sex service was the most significant strategy to reduce the risk of major income reduction (Appendix A).

These behavioral changes and socioeconomic impacts were not found to be associated with changes in condom use frequency, as shown in Table 5. Instead, the increased difficulty of accessing condoms was found to be the most significant determinant of less frequent condom use with an aOR of 10.57 (95% CI 6.60–16.93). Other risk factors of reduced condom use frequency included having unknown or positive HIV status, with aOR of 1.71 (95% CI 1.06–2.75) and 2.40 (95% CI 1.09–5.29), respectively. Cluster analysis also showed that engaging in call sex was associated with reduced condom use frequency (Appendix A).

## 4. Discussion

Our data reveal how the FSW community has been impacted by and has adapted to the COVID-19 pandemic. Most FSWs are severely impacted by the pandemic, indicated by major reductions in client frequency and income, with only a few receiving social supports. Adapting to the risk of COVID-19 transmission, a significant proportion of the FSWs community adopted preventive measures when providing sex services to their clients. Meanwhile, to adapt to social restrictions introduced by the government, some had also taken up online sex work. Concern about COVID-19 infection has driven these behavioral changes. Taking up online sex work was a more likely adaptation among younger FSWs, those with a higher educational level, and those more likely to have access to and proficiency in the required technology. 

Behavioral changes were not found to affect the likelihood of FSWs experiencing reduced client frequency or income. Instead, unemployment and location were identified as predictors of more severe impacts of the pandemic. FSWs in previously tourism-dependent areas were impacted more severely by the pandemic. Evidence has also suggested that client reduction and consequent income reduction may be intentional to some extent as an attempt to prevent themselves from contracting COVID-19. This was indicated by the higher likelihood of major income reduction among FSWs with a higher COVID-19 fear index. 

Regarding HIV, our data showed that the impacts of COVID-19 pandemic, reflected in increased difficulties accessing condoms experienced by some FSWs, were strongly related to reduced condom use frequency. Other than access, attitude toward health risk seemed to be the main factor predicting condom use frequency, since those who reported taking adequate COVID-19-prevention measures also used condoms more frequently. Worryingly, our data also suggest that FSWs with unknown or positive HIV status were more likely to report less frequent condom use during the pandemic.

Our study’s findings about the FSW community’s situation with respect to the pandemic’s socioeconomic impacts confirm the concerns voiced by others. Early in the pandemic, it was noted that key populations, including FSWs, faced a higher risk of socioeconomic impacts compounded by diminished access to social supports [10]. FSWs are especially susceptible due to their work involving close physical contact. Concerns about COVID-19 infection risks may induce FSWs to take fewer clients to the detriment of their socioeconomic stability [9,12]. Our data showed that this concern is a reality with only 36% of FSWs reporting ever receiving any form of social support despite over two-thirds of them experiencing severely reduced income. Similar results were reported by a qualitative study among FSWs in Surakarta, Indonesia [22]. 

At the same time, FSWs also faced greater barriers to accessing healthcare, including HIV-prevention services. In our study, this was indicated by access to condoms. This concern had been raised early in the pandemic [9,12]; another recent study confirmed these concerns with findings similar to ours. FSWs reported various reasons for their reduced condom usage during the pandemic, with the reduced access to supply being one of them [31]. Another potential reason is the more difficult negotiation of condom use with clients [4]. Our data appear not to support this latter factor, however, since client frequency was not associated with condom use frequency.

Access to public services, both social services and healthcare, is the main issue affecting FSWs in this pandemic. Criminalization has often marginalized FSWs to the fringes of society, which inhibits their accessing social supports from either the community or the state. Access to social supports has often been complicated by lack of legal identity among FSWs, which makes qualifying for assistance challenging. Similarly, marginalization has inhibited their access to information about social support and aid provision [32], while their identity as sex workers often isolates them from their families and community [3,7]. 

Meanwhile, healthcare support, which is usually provided in the context of HIV mitigation strategies, may become scarce during the pandemic, with resources being diverted to deal with COVID-19 [12]. Disrupted healthcare access for sex workers during the pandemic has also been reported [24]. Similarly to socioeconomic impacts, this disruption to HIV services has particularly affected migrant sex workers, who have little or no social support in their area [33].

A system of social support organized by the FSW community itself may substitute for the lack of government or larger community support. A scheme of internal resiliency has been shown to be effective in several settings [34,35]. This response has been enabled by an internal social system reinforced by strong social capital existing within an FSW community. Unfortunately, evidence suggests that such a system may not be as robust in the Indonesian FSW community [3,7,8]. Not only would the low extant social capital among Indonesian FSWs impede pandemic relief, but it may also disrupt HIV risk mitigation during the pandemic, including undermining condom use among this group [7,36]. The lack of social capital among FSWs has been exacerbated by the recent Indonesian government policy to disestablish sex workers localizations, such as the famous Dolly Street in Surabaya, which further weakens FSW community resilience [37].

In light of this new circumstance, reports of behavioral changes among FSWs, such as adoption of online sex services or COVID-19 prevention during sex, have emerged. Online sex work is especially relevant, in line with recent developments in sex work that increasingly involve online and virtual services facilitated by near-ubiquitous Internet access [38]. Meanwhile, a series of preventive measures has been recommended to protect FSWs from COVID-19 in instances where in-person service is unavoidable [16]. For HIV, pre-pandemic recommendations, such as consistent condom use and routine HIV tests still stand. However, adhering to these recommendations has become more problematic for FSWs given the disruption of services during the pandemic [24].

Looking at these phenomena through a syndemic lens, we can see how pre-existing socioeconomic factors have affected FSWs’ adoption of these recommended behaviors. For example, our data show that online sex work was more feasible for younger FSWs who had at a least high school education level and live in urban areas. Other facilitators included living alone and being single. Although few studies have attempted to characterize sex workers taking up online work during the pandemic, these characteristics match the profiles of Internet users in Indonesia [39]. This evidence makes clearer the vulnerability of sex workers in Indonesia, with demographic data showing their average lower educational attainment compared to the general population [8]. Similarly, our data showed that condom use during the pandemic was heavily determined by access to condoms and other HIV-preventive services and residential area. Access to condoms is a basic prerequisite for their use [40], and the disrupted supply during the pandemic affects condom use behavior among FSWs [41]. Meanwhile, local responses to the pandemic may also have differently impacted condom use frequency observed in our data as well as by several other studies [5].

Some forms of online sex work are more or less lucrative than others. Our data show that the shift to online sex work does not guarantee an income adequate to support FSWs’ needs; i.e., taking up online sex work was not a determinant of the likelihood of suffering a major income reduction (Table 4 and Appendix A). This result is also noted in other qualitative studies [4,23]. Meanwhile, only around 40% of FSWs were found to adequately adhere to COVID-19 prevention measures during sex work, highlighting their risk of exposure to COVID-19 infection for a significant number of FSWs during the pandemic. Similarly, access to condoms may not on its own ensure condom use during the pandemic. Other socioeconomic and structural factors may likely facilitate or inhibit practice by affecting condom use negotiation or attitude to condom use [3,4].

These results underline the gaps in the provision of a social security net for FSWs. They confirm concerns about the neglect of service provision and protection for FSWs raised by others, both in Indonesia [10] and globally [13,14], and highlight the needs of FSWs for better access to public services, including socioeconomic supports and HIV prevention services. Further, it identifies particularly vulnerable segments of the FSWs community. Older and less educated FSWs, especially those without an alternative source of income, should be prioritized for socioeconomic interventions. Meanwhile, equal access to HIV prevention services should be ensured for all FSWs. Local authorities are best equipped to formulate and implement improved pandemic response and risk mitigation efforts. However, insufficient central government support and resource allocation may prevent effective program implementation at the grassroots level [42].

The strength of this study lies in the large number of FSWs involved. To our knowledge, our study is one of the first large-scale surveys to report on the challenges encountered by FSWs during the pandemic. It quantitatively confirms what has been subjectively expressed by FSWs in previous qualitative studies [4,22,23]. It further identifies specific layers within the FSW community that need prioritized interventions. As vaccine rollout is hindered by misinformation [43] and concerns about the possible subsequent waves of COVID-19 rise, better mitigation plans that accommodate the needs of marginalized populations such as FSWs should be prioritized. 

The study is, however, not without some limitations. Data collection using a self-administered online survey limits our ability to accurately gauge response rates. Another potential issue is that our data describe the situation as it was in the period from September to October 2020. With the developing situation during the pandemic, findings may not reflect more recent changes. In addition, the relatively recent rollout of COVID-19 vaccines, its uptake, and impacts on FSW behavior were not accounted for in this study. Many of our variables also incorporated respondents’ subjective experiences into Likert-scale quantification, which included measurement of changes in FSWs’ client frequency and income level. However, the study’s large sample size may mitigate this subjectivity to some degree.

## 5. Conclusions

Our study supports the recommendations addressing the need to provide continuous socioeconomic supports and HIV prevention services for FSWs. We found that most FSWs experienced serious loss of income, and few had ever received social aid during the COVID-19 pandemic. Some FSWs also found it difficult to access condoms during the pandemic. Adaptive behavioral changes in response to the imposition of social restrictions, such as online sex work, was strongly determined by socioeconomic factors, as well as by awareness of the health risks of in-person sex work. Taking up online sex work, for example, was associated with factors that may influence Internet use, such as younger age and education, as well as with fear of COVID-19. Similarly, access to condoms and HIV-negative status facilitated condom use during the pandemic. These results lead to our recommending more strategic pandemic responses targeting FSWs, encompassing equal access to HIV prevention services as well as targeted socioeconomic supports to especially vulnerable segments of the FSW community.

## Figures and Tables

**Table 1 ijerph-19-01361-t001:** Demographic and socioeconomic characteristics.

Variables (n = 951)	
Location, n (%)	
Greater Jakarta	334 (35.1)
Bandung	250 (26.3)
Yogyakarta	152 (16.0)
Bali	215 (22.6)
Age (years), median (IQR)	26 (23–31)
Education, n (%)	
Not completed high school	430 (45.2)
High school	480 (50.5)
College degree	41 (4.3)
Employment other than sex work, n (%)	
None	606 (63.7)
Employed	345 (36.3)
Marital status, n (%)	
Single	388 (40.8)
Cohabitating	49 (5.2)
Married	148 (15.6)
Widowed/divorced	366 (38.5)
Housing, n (%)	
Alone	355 (37.3)
With room mate	200 (21.0)
With family	396 (41.6)
HIV status, n (%)	
Negative	488 (51.3)
Unknown	376 (39.5)
Positive	87 (9.1)
Fear of COVID-19 score	
Low	493 (51.8)
High	458 (48.2)
Sanitation and personal hygiene during sex work, n (%)	
Low adherence	477 (50.2)
High adherence	474 (49.8)
PPE use during sex work, n (%)	
Low adherence	366 (38.5)
High adherence	585 (61.5)
Engaged in online sex work, n (%)	
No	605 (63.8)
Yes	346 (36.4)
Client frequency changes, n (%)	
Little to no effect	306 (32.2)
Major reduction	645 (67.8)
Income reduction, n (%)	
Little to no reduction	275 (28.9)
Major reduction	676 (71.1)
Received social support, n (%)	
Never	606 (63.7)
At least once	345 (36.3)
Changes to condom access, n (%)	
No change or easier	792 (83.3)
More difficult	159 (16.7)
Changes to condom use, n (%)	
No change or more often	832 (87.5)
Less often	119 (12.5)

**Table 2 ijerph-19-01361-t002:** Determinants of taking up online sex work.

Variables (n = 951)	Online Sex Work	*p* Value	*p* Value
	(n = 346)	cOR (95% CI)	aOR (95% CI)
Location, n (%)		<0.001 **	0.002 **
Bandung (n = 250)	87 (34.8)	Ref.	Ref.
Greater Jakarta (n = 334)	152 (45.5)	1.57 (1.12–2.19)	1.27 (0.87–1.84)
Yogyakarta (n = 152)	49 (32.2)	0.89 (0.58–1.37)	0.69 (0.43–1.12)
Bali (n = 215)	58 (27.0)	0.69 (0.47–1.03)	0.64 (0.41–0.98)
Age (years)	25 (22–29)	<0.001 ** 0.95 (0.93–0.97)	0.002 ** 0.96 (0.93–0.98)
Education		<0.001 **	
High school not completed (n = 430)	114 (26.5)	Ref.	Ref.
High school (n = 480)	212 (44.2)	2.19 (1.66–2.90)	2.00 (1.48–2.70)
College degree (n = 41)	20 (48.8)	2.64 (1.38–5.05)	2.38 (1.18–4.79)
Employment other than sex work		<0.001 **	<0.001 **
None (n = 606)	180 (29.7)	Ref.	Ref.
Employed (n = 345)	166 (48.1)	2.20 (1.67–2.89)	2.22 (1.63–3.01)
Marital status, n (%)		0.132	0.095
Single (n = 388)	176 (45.4)	Ref.	Ref.
Cohabitating (n = 49)	15 (30.6)	0.53 (0.28–1.01)	0.77 (0.38–1.55)
Married (n = 148)	58 (39.2)	0.78 (0.53–1.14)	1.34 (0.83–2.16)
Widowed/divorced (n = 366)	97 (26.5)	0.43 (0.32–0.59)	0.78 (0.54–1.13)
Housing, n (%)		0.007 **	0.009 **
Alone (n = 355)	146 (41.1)	Ref.	Ref.
With roommate (n = 200)	66 (33.0)	0.71 (0.49–1.01)	0.65 (0.43–0.97)
With family (n = 396)	134 (33.8)	0.73 (0.54–0.99)	0.60 (0.42–0.85)
HIV status, n (%)		0.543	-
Negative (n = 488)	172 (35.2)	Ref.	
Unknown (n = 376)	146 (38.6)	1.17 (0.88–1.54)	
Positive (n = 87)	28 (32.2)	0.87 (0.54–1.42)	
Fear of COVID-19 score		<0.001 **	<0.001 **
Low (n = 493)	154 (31.2)	Ref.	Ref.
High (n = 458)	192 (41.9)	1.59 (1.22–2.07)	1.69 (1.27–2.25)

* *p* < 0.05; ** *p* < 0.01; Ref.: reference group.

**Table 3 ijerph-19-01361-t003:** Determinants of adherence to COVID-19 prevention protocol for sex work.

Variables (n = 951)	Sanitation and Personal Hygiene (n = 474)			PPE Use During Sex Work (n = 585)		
	n (%)	*p* Value cOR (95% CI)	*p* Value aOR (95% CI)	n (%)	*p* Value cOR (95% CI)	*p* Value aOR (95% CI)
Location, n (%)		<0.001 **	0.003 **		0.224	0.199
Bandung (n = 250)	117 (46.8)	Ref.	Ref.	155 (62.0)	Ref.	Ref.
Greater Jakarta (n = 334)	146 (43.7)	0.88 (0.64–1.23)	0.79 (0.55–1.13)	216 (64.7)	0.51 (1.12–1.58)	0.91 (0.63–1.32)
Yogyakarta (n = 152)	98 (64.5)	2.06 (1.36–3.12)	1.68 (1.07–2.64)	94 (61.8)	0.99 (0.66–1.50)	0.87 (0.56–1.37)
Bali (n = 215)	113 (52.6)	1.26 (0.87–1.81)	1.17 (0.78–1.76)	120 (55.8)	0.77 (0.53–1.12)	0.66 (0.44–0.98)
Age (years)	25 (22–30)	0.012 * 1.03 (1.01–1.04)	0.183 1.02 (0.99–1.04)	25 (23–30)	0.078 0.98 (0.96–1.00)	0.264 0.99 (0.96–1.01)
Education		0.477	-		0.031 *	0.035 *
Not completed high school (n = 430)	220 (51.2)	Ref.		255 (59.3)	Ref.	Ref.
High school (n = 480)	237 (49.4)	0.93 (0.72–1.21)		311 (64.8)	1.26 (0.96–1.65)	1.17 (0.88–1.55)
College degree (n = 41)	17 (41.5)	0.68 (0.35–1.29)		19 (46.3)	0.59 (0.31–1.13)	0.50 (0.26–0.98)
Employment other than sex work		0.031 *	0.189		0.056	0.077
None (n = 606)	286 (47.2)	Ref.	Ref.	359 (59.2)	Ref.	Ref.
Employed (n = 345)	188 (54.5)	1.34 (1.03–1.75)	1.21 (0.91–1.62)	226 (65.5)	1.31 (0.99–1.72)	1.31 (0.97–1.76)
Marital status, n (%)		0.102	0.164		0.019 *	0.206
Single (n = 388)	193 (49.7)	Ref.	Ref.	255 (65.7)	Ref.	Ref.
Cohabitating (n = 49)	16 (32.7)	0.49 (0.26–0.92)	0.47 (0.24–0.92)	29 (59.2)	0.76 (0.41–1.39)	0.95 (0.50–1.81)
Married (n = 148)	75 (50.7)	1.04 (0.71–1.52)	0.79 (0.50–1.25)	98 (66.2)	1.02 (0.69–1.53)	1.19 (0.75–1.90)
Widowed/divorced (n = 366)	190 (51.9)	1.09 (0.82–1.45)	0.91 (0.65–1.28)	203 (55.5)	0.65 (0.48–0.87)	0.78 (0.55–1.11)
Housing, n (%)		0.039 *	0.219		0.029 *	0.10 *
Alone (n = 355)	188 (53.0)	Ref.	Ref.	234 (65.9)	Ref.	Ref.
With roommate (n = 200)	84 (42.0)	0.64 (0.45–0.91)	0.71 (0.48–1.05)	109 (54.5)	0.62 (0.44–0.88)	0.56 (0.38–0.82)
With family (n = 396)	202 (51.0)	0.93 (0.69–1.23)	0.91 (0.66–1.27)	242 (61.1)	0.81 (0.60–1.20)	0.72 (0.52–1.01)
HIV status, n (%)		<0.001 **	<0.001 **		0.709	-
Negative (n = 488)	275 (56.4)	Ref.	Ref.	301 (61.7)	Ref.	
Unknown (n = 376)	141 (37.5)	0.47 (0.35–0.61)	0.47 (0.36–0.63)	234 (62.2)	1.02 (0.78–1.35)	
Positive (n = 87)	58 (66.7)	1.55 (0.96–2.50)	1.41 (0.86–2.33)	50 (57.5)	0.84 (0.53–1.33)	
Fear of COVID-19 score		0.001 **	0.007 **		0.010 *	0.015 *
Low (n = 493)	219 (44.4)	Ref.	Ref.	284 (57.6)	Ref.	Ref.
High (n = 458)	255 (55.7)	1.57 (1.22–2.03)	1.44 (1.10–1.89)	301 (65.7)	1.41 (1.09–1.84)	1.40 (1.07–1.83)

* *p* < 0.05; ** *p* < 0.01; Ref.: reference group.

**Table 4 ijerph-19-01361-t004:** Risk factors for major reduction of income during the COVID-19 pandemic.

Variables (n = 951)	Major Income Reduction (n = 676)	*p* Value cOR (95% CI)	*p* Value aOR (95% CI)
Location, n (%)		<0.001 **	0.004 **
Bandung (n = 250)	170 (68.0)	Ref.	Ref.
Greater Jakarta (n = 334)	220 (65.9)	0.91 (0.64–1.29)	1.07 (0.72–1.59)
Yogyakarta (n = 152)	130 (85.5)	2.78 (1.65–4.70)	2.78 (1.54–5.00)
Bali (n = 215)	156 (72.6)	1.24 (0.83–1.86)	1.31 (0.84–2.04)
Age (years)	27 (22–30)	0.006 ** 1.03 (1.01–1.05)	0.133 1.02 (0.99–1.05)
Education		0.349	-
Not completed high school (n = 430)	307 (71.4)	Ref.	
High school (n = 480)	344 (71.7)	1.01 (0.76–1.35)	
College degree (n = 41)	25 (61.0)	0.63 (0.32–1.21)	
Employment other than sex work		0.016 *	0.004 **
None (n = 606)	447 (73.8)	Ref.	Ref.
Employed (n = 345)	229 (66.4)	0.70 (0.53–0.94)	0.61 (0.44–0.86)
Marital status, n (%)		0.123	0.376
Single (n = 388)	261 (67.3)	Ref.	Ref.
Cohabitating (n = 49)	37 (75.5)	1.50 (0.76–2.98)	1.81 (0.85–3.84)
Married (n = 148)	104 (70.3)	1.15 (0.76–1.74)	0.87 (0.53–1.43)
Widowed/divorced (n = 366)	274 (74.9)	1.45 (1.06–1.99)	0.99 (0.67–1.47)
Housing		0.658	-
Alone (n = 355)	254 (71.5)	Ref.	
With roommate (n = 200)	137 (68.5)	0.87 (0.59–1.26)	
With family (n = 396)	285 (72.0)	1.02 (0.74–1.40)	
HIV status		<0.001 **	<0.001 **
Negative (n = 488)	380 (77.9)	Ref.	Ref.
Unknown (n = 376)	251 (66.8)	0.57 (0.42–0.77)	0.58 (0.42–0.82)
Positive (n = 87)	45 (51.7)	0.31 (0.19–0.49)	0.20 (0.12–0.34)
Fear of COVID-19 score		<0.001 **	<0.001 **
Low (n = 493)	318 (64.5)	Ref.	Ref.
High (n = 458)	358 (78.2)	1.97 (0.48–2.63)	1.88 (1.36–2.59)
Engaged in online sex work		0.055	0.165
No (n = 605)	433 (73.2)	Ref.	Ref.
Yes (n = 346)	233 (67.3)	0.75 (0.57–1.01)	0.79 (0.57–1.10)
Sanitation & personal hygiene during sex work, n (%)		0.001 **	0.048 *
Low adherence (n = 477)	316 (66.2)	Ref.	Ref.
High adherence (n = 474)	360 (75.9)	1.61 (1.21–2.14)	1.39 (1.00–1.91)
PPE use during sex work, n (%)		0.642	-
Low adherence (n = 366)	257 (70.2)	Ref.	
High adherence (n = 585)	419 (71.6)	1.07 (0.80–1.43)	
Client frequency		<0.001 **	<0.001 **
Little to no change (n = 306)	156 (51.0)	Ref.	Ref.
Major reduction (n = 645)	520 (80.6)	4.00 (2.97–5.39)	3.32 (2.41–4.56)

* *p* < 0.05; ** *p* < 0.01; Ref.: reference group.

**Table 5 ijerph-19-01361-t005:** Risk factors for less frequent condom use during COVID-19 pandemic.

Variables (n = 951)	Less Frequent Condom Use (n = 119)	*p* Value cOR (95% CI)	*p* Value aOR (95% CI)
Location, n (%)		0.064	0.062
Bandung (n = 250)	37 (14.8)	Ref.	Ref.
Greater Jakarta (n = 334)	46 (13.8)	0.92 (0.58–1.47)	0.63 (0.36–1.10)
Yogyakarta (n = 152)	9 (5.9)	0.36 (0.17–0.77)	0.33 (0.14–0.77)
Bali (n = 215)	27 (12.6)	0.83 (0.49–1.41)	0.61 (0.32–1.15)
Age (years)		<0.001 ** 0.94 (0.91–0.97)	0.955 1.00 (0.96–1.04)
Education		0.341	-
Not completed high school (n = 430)	54 (12.6)	Ref.	
High school (n = 480)	63 (13.1)	1.05 (0.71–1.55)	
College degree (n = 41)	2 (4.9)	0.36 (0.08–1.52)	
Employment other than sex work		0.709	-
None (n = 606)	74 (12.2)	Ref.	
Employed (n = 345)	45 (13.0)	1.08 (0.73–1.60)	
Marital status, n (%)		<0.001 **	0.048 *
Single (n = 388)	66 (17.0)	Ref.	Ref.
Cohabitating (n = 49)	8 (16.3)	0.95 (0.43–2.12)	0.83 (0.32–2.16)
Married (n = 148)	22 (14.9)	0.85 (0.50–1.44)	0.76 (0.39–1.46)
Widowed/divorced (n = 366)	23 (6.3)	0.33 (0.20–0.54)	0.42 (0.23–0.77)
Housing		0.307	-
Alone (n = 355)	52 (14.6)	Ref.	
With roommate (n = 200)	22 (11.0)	0.72 (0.42–1.23)	
With family (n = 396)	45 (11.4)	0.75 (0.49–1.15)	
HIV status		0.006 **	0.026 *
Negative (n = 488)	45 (9.2)	Ref.	Ref.
Unknown (n = 376)	62 (16.5)	1.94 (1.29–2.93)	1.71 (1.06–2.75)
Positive (n = 87)	12 (13.8)	1.58 (0.80–3.12)	2.40 (1.09–5.29)
Fear of COVID-19 score		0.469	-
Low (n = 493)	58 (11.8)	Ref.	
High (n = 458)	61 (13.3)	1.15 (0.79–1.69)	
Engaged in online sex work		0.001 **	0.240
No (n = 605)	59 (9.8)	Ref.	Ref.
Yes (n = 346)	60 (17.3)	1.94 (1.32–2.86)	1.32 (0.83–2.08)
Sanitation and personal hygiene during sex work, n (%)		0.001 **	<0.001 **
Low adherence (n = 477)	82 (17.2)	Ref.	Ref.
High adherence (n = 474)	37 (7.8)	0.41 (0.27–0.62)	0.42 (0.26–0.68)
PPE use during sex work, n (%)		0.031 *	0.475
Low adherence (n = 366)	35 (9.6)	Ref.	Ref.
High adherence (n = 585)	84 (14.4)	1.59 (1.04–2.41)	1.20 (0.73–1.98)
Condom access		<0.001 **	<0.001 **
Same or easier (n = 792)	51 (6.4)	Ref.	Ref.
More difficult (n = 159)	68 (42.8)	10.86 (7.11–16.58)	10.57 (6.60–16.93)
Client frequency		0.787	-
Little to no change (n = 306)	37 (12.1)	Ref.	
Major reduction (n = 645)	82 (12.7)	1.06 (0.70–1.60)	
Income change		0.731	-
Little to no change (n = 275)	36 (13.1)	Ref.	
Major reduction (n = 71.1)	83 (12.3)	0.93 (0.61–1.41)	

* *p* < 0.05; ** *p* < 0.01; Ref.: reference group.

## Data Availability

The data presented in this study is available upon reasonable request and unanimous consent of all investigators involved.

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
