# Peer review of "Behavioral Changes, Adaptation, and Supports among Indonesian Female Sex Workers Facing Dual Risk of COVID-19 and HIV in a Pandemic"

_ijerph, 2022, doi:10.3390/ijerph19031361_

Round 1

Reviewer 1 Report

The authors present a timely investigation of FSW in Indonesia in the context of COVID-19.

I would like to hear more about the online sex work measure. What kind of sex work is meant here? Onlyfans? Could it compensate for offline work? I think it would be necessary here to run a cluster analysis that investigates which behaviors/experiences are shown jointly to carve out a more clear profile of who is at risk

I am concerned about the dichotomization of the COVID prevention measure. The instrument uses a Likert-type scale and not dichotomous yes/no questions as an answer format. This scale cannot be simply dichotomized, the cut-off point at 66% is at random and not based on any standardization. No distribution information for the scale are provided. Especially since the measures assessed are not validated in regard to their protective potential, dichotomizing the scale puts participants at random into a low/high category while they may be at 65 and 67% and hence very similar. My strong suggestion is to use the scale as a continuous predictor, or, if that is not deemed feasible, to provide full information on the variance and distribution of the scale items, and then use at least quartiles or any form of data reduction that still does justice to the character of the scale.

I am not sure how the univariable and multivariable regressions where computed? Usually only certain predictors below a threshold are included in the multivariable regression, here it seems all are included.

I would recommend to have the MS proof read by a native speaker to iron out a number of grammar and spelling errors

Author Response

  1. I would like to hear more about the online sex work measure. What kind of sex work is meant here? Onlyfans? Could it compensate for offline work?

Response:

Thank you for your interest in this topic. We have added more explanation regarding online sex work referred in our manuscript (please refer to lines 153-158). Please note that adult content making (such as Onlyfans) is not mainstream among Indonesian FSWs and would only be accessible to the most internet-savvy ones as it requires use of VPN.

  1. I think it would be necessary here to run a cluster analysis that investigates which behaviors/experiences are shown jointly to carve out a more clear profile of who is at risk

Response:

We agree that it is interesting to see how each type of online sex work measured would correlates with behavioral outcomes in our study. Accordingly, we conducted a couple of logistic regression using each type of online sex work (video call sex, call sex, and chat sex/sexting services) as standalone independent variables. The result is shown in supplementary Table S2.

  1. I am concerned about the dichotomization of the COVID prevention measure. The instrument uses a Likert-type scale and not dichotomous yes/no questions as an answer format. This scale cannot be simply dichotomized, the cut-off point at 66% is at random and not based on any standardization. No distribution information for the scale are provided. Especially since the measures assessed are not validated in regard to their protective potential, dichotomizing the scale puts participants at random into a low/high category while they may be at 65 and 67% and hence very similar. My strong suggestion is to use the scale as a continuous predictor, or, if that is not deemed feasible, to provide full information on the variance and distribution of the scale items, and then use at least quartiles or any form of data reduction that still does justice to the character of the scale.

Response:

Thank you for your input. We apologize for the lack of description on the numerical score. We have added data distribution information as well as internal reliability information (Cronbach α value) in Table S1. Our initial intention for using 66.67% as cutoff point were meant to represent respondents who at least answered ‘often’ (corresponding to score 2 out of 3) to all behavioral items in the measure. However, we understand that it may seems arbitrary considering the score was not a validated one. However, for the same reason we don’t believe using the numerical score as dependent variable would be appropriate either. As such, we have switched to using median value (60.61%) as the cutoff point for dividing high adherence and low adherence groups. Please refer to lines 159-165 and changes made to Table S1.

  1. I am not sure how the univariable and multivariable regressions where computed? Usually only certain predictors below a threshold are included in the multivariable regression, here it seems all are included.

Response:

Thank you for your input. We understand that the convention for multivariate regression is only involving independent variables with p < 0.25 in bivariate analysis. Our initial intention was, as this is a somewhat exploratory study, we involved all independent variables in multivariate regression. The result seems to justify the convention, however, as none of the variables that failed to pass that threshold was found to be significant independent determinants in the multivariate regression. As such, we agree with your suggestion and redo the regression analyses according to the convention with streamline data presentation.

  1. I would recommend to have the MS proof read by a native speaker to iron out a number of grammar and spelling errors

Response:

Thank you for your suggestion. We have run the latest draft with a native speaker colleague of ours.

Reviewer 2 Report

This article uses a cross-sectional survey to assess behavioral changes, adaptation, and support among Indonesian female sex workers (FSWs) who face the risks of COVID-19 and HIV during a pandemic.  A secondary objective is to clarify factors influencing socioeconomic impacts and behavioral changes in response to the COVID-19 pandemic among Indonesian FSWs. Authors assert a need for this study due to the fact the sex workers are particularly susceptible to contacting COVID-19 and HIV during the COVID-19 pandemic while often being marginalized in society, which can impede access to healthcare services.

Feedback is as follows:

  1. Lines 15-16 – Check wording and verb tense in “Respondents was incentivized”.
  2. Lines 32-33 – In the sentence “In September 2021, it has accumulated over 224 million infection and over 4.6 million mortalities”, author should specify the context of these COVID-19 numbers. Are these infection and mortalities at the global level?  If so, authors should state this.
  3. Overall, the Introduction should be expanded with more discussion on the sociocultural, economic, and demographic factors that make FSWs in Indonesia more susceptible to COVID-19 and HIV.
  4. Lines 44-45 – Check verb tense in “There were concerns” and consider using present tense (are).
  5. Line 46- What is considered ‘riskier sex behavior’? Does this mean unprotected sex?  The authors should clarify.
  6. Line 52 – For “was envisioned”, consider saying ‘has been envisioned’.
  7. Lines 55-56- It would be helpful to get more insight into the extent of HIV preventions services for FSWs in brothels. What do these services entail?  For example, do the services include prevention, counseling, testing, referrals?  Authors should expand on this point.
  8. For the study design, what were the specifics on the survey in terms of length/duration to complete, number and type of items, and language used?
  9. Did an online survey present any access issues for respondents?
  10. What was the response rate for the survey?
  11. Line 113 – For the recruitment of participants, what are considered FSW “target areas”? The authors should expand on this statement.
  12. Is there any insight into the income differential between the online sex work and the in-person work?
  13. Lines 200-201 – Check wording in “Another risk factors of reduced”.
  14. Line 338 – for the recommendation for an improved pandemic response for FSWs, at what level would this take place (for example, national, local)? Authors should expand on this statement.
  15. The paper should be reviewed for English language and style.

Overall, this is an insightful, pertinent, and unique study on a very important topic.  It is interesting to read.  Attending to some clarifying questions may help to improve the overall paper.

Author Response

  1. Lines 15-16 – Check wording and verb tense in “Respondents was incentivized”.

Response:

Thank you for your input. We have edited the sentence with input from a native speaker colleague.

  1. Lines 32-33 – In the sentence “In September 2021, it has accumulated over 224 million infection and over 4.6 million mortalities”, author should specify the context of these COVID-19 numbers. Are these infection and mortalities at the global level? If so, authors should state this.

Response:

Thank you for noticing. We agree that the initial version was unclear. We have edited to sentence accordingly to clarify that the figure was meant globally (please refer to lines 33-34).

  1. Overall, the Introduction should be expanded with more discussion on the sociocultural, economic, and demographic factors that make FSWs in Indonesia more susceptible to COVID-19 and HIV.

Response:

Thank you for your suggestion. We have added this discussion in the introduction section. Please refer to lines 40-60.

  1. Lines 44-45 – Check verb tense in “There were concerns” and consider using present tense (are).

Response:

Thank you for your input. This sentence was abrogated during the revision process. We take note on your input to fix other possibly similar errors.

  1. Line 46- What is considered ‘riskier sex behavior’? Does this mean unprotected sex? The authors should clarify.

Response:

Thank you for your suggestion. We have clarified this phrase. Please refer to lines 48-49 of the revised manuscript.

  1. Line 52 – For “was envisioned”, consider saying ‘has been envisioned’.

Response:

Thank you for your suggestion. This phrase has been revised. Please refer to lines 83 of the revised manuscript.

  1. Lines 55-56- It would be helpful to get more insight into the extent of HIV preventions services for FSWs in brothels. What do these services entail? For example, do the services include prevention, counseling, testing, referrals?  Authors should expand on this point.

Response:

Thank you for your input. We have to clarify that the brothels themselves often times do not provide these services. However, their role as localization of FSW activities help healthcare providers, both public and private, to reach FSW community and provide these services for them. The services provided usually included free condom distribution and routine mobile HIV test. This information has been added to the text. Please refer to lines 74-80.

  1. For the study design, what were the specifics on the survey in terms of length/duration to complete, number and type of items, and language used? Did an online survey present any access issues for respondents? What was the response rate for the survey?

Response:

Thank you for your suggestion. We bundled the response to these three queries as they are closely related to the survey methodology. We have added more detail to the survey design, including language used, number and types of items, and duration required to complete (please refer to lines 128-133). Regarding access, collaborating organizations who help distributed the recruitment material also provided assistance for potential respondents who faced difficulties accessing the survey. It has helped us mitigate the access issue during data collection. This information has been added to the text (please refer to lines 138-143). Unfortunately, our online survey methodology does not provide us with a figure of how many people was reached by our survey recruitment materials. As such, we can’t provide response rate figure. This fact has been added to the limitation section of the manuscript (lines 377-378).

  1. Line 113 – For the recruitment of participants, what are considered FSW “target areas”? The authors should expand on this statement.

Response:

Thank you for your input. The ‘target areas’ phares is meant to refer to the four cities where the study is conducted which includes Jakarta, Bandung, Yogyakarta, and Bali. It does not mean to refer to specific areas within these cities. This phrase has been clarified in the text. Please refer to lines 134-135.

  1. Is there any insight into the income differential between the online sex work and the in-person work?

Response:

Thank you for your insightful query. This is an interesting question to answer. Unfortunately, our study did not collect the data for this specific question. It would be interesting to conduct a more specific study on online sex work phenomenon in Indonesia in the future. The lack of insight regarding objective income differential between online and in-person sex work has been noted in the limitation section regarding subjectivity of some measures used in this study (lines 383-385).

  1. Lines 200-201 – Check wording in “Another risk factors of reduced”.

Response:

Thank you for your suggestion. However, this sentence was abrogated during revision process and is no longer relevant.

  1. Line 338 – for the recommendation for an improved pandemic response for FSWs, at what level would this take place (for example, national, local)? Authors should expand on this statement.

Response:

Thank you for your input. We agree that this is an interesting and important point in our manuscript. We believe that local health authorities are the best positioned to plan and implement direct tangible improvement regarding pandemic response and HIV mitigation for FSW community. However, our experience shown that local authorities can only provide such response if they receive logistical, financial, and regulatory support from the central government. Lack of such support has been cited as barriers to HIV program implementation in the past. We have added this expansion to the manuscript (please refer to lines 365-368).

  1. The paper should be reviewed for English language and style.

Response:

Thank you for your suggestion. We have run the revised manuscript through a native speaker colleague of ours to improve its readability.

Round 2

Reviewer 1 Report

I have seen a previous version of the MS and I agree that the current version has improved. My major concerns are alleviated, but one still remains: the dichotomization of the COVID-19 protection measure scale. I have problems with the scale in general and with its use. The scale is a continuous Likert type scale and dichotomizing it leads to data loss and potentially biased interpretations. The median split is an option, but still suboptimal - to make it work a more careful analysis is necessary. A median split rests on the assumption that all items of the scale are equal in terms of contribution to the external validity of the scale and its internal reliability, and in terms of COVID-19 protection potential. By simply median splitting the scale there is the possibility that a sex worker who engaged in the most efficient measures gets sorted into the "low adherence" category while she actually engaged in the most efficient (use of mask, face shield, no kissing) ones, and refrained from less relevant ones (shower after sex). Given the novelty of the scale, I would suggest that the authors provide a more robust analysis. That is (1) conduct a Factor analysis to determine if there are underlying factors (more or less efficient measures) and provide an internal reliability with Alpha when item deleted statistic. That way a better interpretation of the scale would be possible and a better split (potentially on subscales, along a median) can be achieved.

Author Response

Response to Reviewer #1

  1. Given the novelty of the scale, I would suggest that the authors provide a more robust analysis. That is (1) conduct a Factor analysis to determine if there are underlying factors (more or less efficient measures) and provide an internal reliability with Alpha when item deleted statistic. That way a better interpretation of the scale would be possible and a better split (potentially on subscales, along a median) can be achieved.

Response:

Thank you for your acknowledgement of our previous revision. We did not conduct factor analysis in initial draft as we did not see validation of the measure to be a primary objective of the study. However, we took this input to heart and conduct factor analysis as suggested. The result found that 2 out of 11 original items did not load significantly to the final measure and was dropped. The remaining 9 items was divided into two subscales: 1) sanitation and personal hygiene (6 items), and 2) PPE usage during sex work (3 items). One item (mask-wearing during sex work) was found to cross-load between the two factors but was finally included in the PPE-use subscale due to stronger loading factor. The complete result of factor analysis can be seen in changes made to Table S1. We also made relevant changes to the method, result, and discussion changes to reflect the change made to this variable.

Reviewer 2 Report

The authors have done well to respond to the suggested feedback.  The paper is clearer and improved. 

Author Response

Response to Reviewer #2

  1. The authors have done well to respond to the suggested feedback. The paper is clearer and improved.

Response:

Thank you very much for your acknowledgement and appreciation.